# Clarifying the Heterogeneity in Response to Vitamin D in the Development, Prevention, and Treatment of Type 2 Diabetes Mellitus: A Narrative Review

**DOI:** 10.3390/ijerph20126187

**Published:** 2023-06-20

**Authors:** Jacob M. Hands, Patrick G. Corr, Leigh A. Frame

**Affiliations:** The George Washington University School of Medicine & Health Sciences, Washington, DC 20037, USA

**Keywords:** vitamin D, type 2 diabetes mellitus, bioavailability, sex-specific response, autoimmunity, health behaviors, chronic diseases, public health

## Abstract

In this review, we explore the potential drivers of heterogeneity in response to Vitamin D (VitD) therapy, such as bioavailability, sex-specific response, and autoimmune pathology, in those at risk for and diagnosed with T2DM. In addition, we propose distinct populations for future interventions with VitD. The literature concerning VitD supplementation in the prevention, treatment, and remission of type 2 diabetes mellitus (T2DM) spans decades, is complex, and is often contradictory with mixed findings upon intervention. By association, VitD status is powerfully predictive with deficient subjects reporting greater risk for T2DM, conversion to T2DM from prediabetes, and enhanced response to VitD therapy. Preclinical models strongly favor intervention with VitD owing to the pleiotropic influence of VitD on multiple systems. Additional research is crucial as there remain many questions unanswered that are related to VitD status and conditions such as T2DM. Future research must be conducted to better understand the potentially spurious relationships between VitD status, supplementation, sun exposure, health behaviors, and the diagnosis and management of T2DM. Public health practice can greatly benefit from a better understanding of the mechanisms by which we can reliably increase VitD status and how this can be used to develop education and improve health behaviors.

## 1. Introduction

The literature concerning vitamin D (VitD) supplementation in the prevention, treatment, and remission of type 2 diabetes mellitus (T2DM) spans decades, is complex, and is often contradictory with authors reporting mixed findings upon intervention with ergocalciferol (VitD2) or cholecalciferol (VitD3) (either alone or in combination with other agents such as calcium and fish oil [1,2,3,4]). This is also true of the epidemiological literature, including a negative Mendelian randomization analysis in a Chinese population [5,6,7]. There likely exist a litany of factors contributing to this heterogeneity, such as trial design, baseline VitD status, variability in VitD metabolism, ethnicity, administered VitD dose, BMI, and nutritional status. Collectively, these forces may influence subject response to therapy and thus moderate clinical outcomes.

By association, serum VitD status is powerfully predictive in the setting of numerous conditions, especially T2DM, with deficient subjects (frank deficiency is typically defined as <50 nmol/dL or <20 ng/dL) reporting a greater risk for T2DM, conversion to T2DM from prediabetes, and an enhanced response to VitD therapy [4,8,9]. The National Academy of Medicine (formerly the Institute of Medicine) and the Endocrine Society have both set their definition of VitD deficiency (and insufficiency) based on the effects of serum 25(OH)D concentration on bone health, not on non-skeletal effects such as immune competence or the prevention of chronic disease, which require a much higher VitD status [10]. In accordance with these guidelines, most studies currently use less than 20 ng/mL (50 nmol/L) for VitD deficiency and less than 30 ng/mL (75 nmol/L) or 32 ng/mL (80 nmol/L) for VitD insufficiency; however, it has been suggested that serum 25(OH)D concentrations of 40 ng/mL (100 nmol/L) or even 48 ng/mL (120 nmol/L) are necessary for some of the extra-skeletal health benefits, such as prevention of chronic disease and the promotion of immune competence. In fact, many in the VitD field are recommending an optimal range of 40–60 ng/mL (100–150 nmol/L), which is more in line with what is seen in free living individuals in equatorial regions living a traditional lifestyle and with high melanin concentration [11,12].

The VitD status necessary to prevent T2DM is likely substantially larger than that for bone health—and thus the current National Academy of Medicine and Endocrine Society guidelines—but the optimal serum 25(OH)D is not clear. In a randomized, placebo-controlled trial of weekly 25,000 IU of VitD3, Zaromytidou et al. found increasing VitD status to 28.71 ± 9.03 ng/mL (71.78 ± 22.58 nmol/L) at 12 months showed an improvement in fasting glucose and glycated hemoglobin (A1C) [13]. In another randomized, placebo-controlled study (4000 IU VitD3 daily), von Hurst et al. showed that increases in 25(OH)D of 32+ ng/mL (80+ nmol/L) resulted in an improvement in insulin sensitivity and resistance (HOMA2/HOMA2-IR) and fasting glucose in that cohort [14]. However, some key biomarkers related to preventing chronic diseases such as T2DM including C-peptide, lipid profile, and high sensitivity C-reactive protein (CRP) did not improve, suggesting higher VitD status/exposure may be necessary to affect these markers and, perhaps, ultimately to prevent the disease.

Moreover, preclinical models strongly favor intervention with VitD owing to the pleiotropic influence of VitD on multiple systems, including its impact on islet beta cell homeostasis and performance, GLUT transporter upregulation, endogenous antioxidant stimulation, and fatty acid receptor modulation in concert with its primary role in the maintenance of the calcium-phosphate economy and bone mineralization [1,15].

In the aftermath of competing T2DM meta-analysis documenting heterogeneity in the randomized control trial participants that were supplemented with VitD, the question of which sub-populations may derive benefit from treatment must be at the forefront of future interventions. In this review, we explore potential drivers of heterogeneity in response to VitD therapy, such as bioavailability, sex-specific response, autoimmune pathology, and oxidative stress, in those at risk for and diagnosed with T2DM; in addition, we propose distinct populations for future interventions with VitD.

## 2. Bioavailability and Response with Supplementation

### 2.1. Vitamin D Metabolism

VitD status is chiefly regulated through photoproduction during exposure to ultraviolet-B (UVB) radiation from the sun. Within the epidermis, this UVB exposure induces the conversion of 7-dehydrocholesterol, a sterol abundant in the epidermis, to form VitD through a non-enzymatic, photolytic isomerization process, which is estimated to contribute to 80–90% of the total serum VitD concentration (Figure 1) [16,17,18]. Importantly, VitD photoproduction occurs with suberythemal (non-inflammatory) doses and over exposure (inflammatory) actually leads to the destruction of VitD in the skin [19]. Therefore, VitD photoproduction can be obtained without increasing the risk for skin cancer and may actually reduce the risk of melanoma [20]. However, the exact exposure time that is needed depends on factors related to the environment (latitude, season, time of day, altitude) and to the individuals (amount of skin exposed, melanin concentration in the skin, sunscreen) [20,21,22]. Leaders in the field have worked to develop a smartphone app to allow the tracking of VitD status, including the recommendation/monitoring of sensible sun exposure to allow for VitD photoproduction in suberythemal doses [23].

In addition to baseline UVB exposure, serum VitD status is also influenced by the dietary intake of VitD—with each additional intake of 100 IU reflecting a 2.0–5.0 ng/dL (5 nmol/L) increase in serum concentration, according to a meta-analysis [24]. The recommended daily allowance (RDA) for VitD intake in the United States is currently set to 600–800 IU (~15 mcg per day). It should be noted that an RDA of 800 IU, or serum 25(OH)D concentrations of about <30 ng/dL (75 nmol/L), is likely to produce an insufficient VitD status, but is typically sufficient to prevent frank deficiency. To achieve sufficiency (30+ ng/dL, 75+ nmol/L), it is estimated that a dietary intake of 1500–2000 IU (~50 mcg) would be required for an average person [10,24,25]. However, there is large interindividual variation in the need for VitD, which is dependent upon lifestyle (sun exposure, diet, supplementation), digestion (bioaccessibility and absorption), and nutritional requirements. In the absence of supplementation or enhanced exposure to full body UVB (20+ min per day, depending on the melanin concentration of skin), achieving VitD sufficiency is challenging with several authors reporting that >50% of the world’s population remains in insufficient supply [24].

Once converted into VitD, approximately 85% is shuttled throughout the body by the VitD binding protein (VDBP), an a2-globulin, and about 15% by albumin. Free VitD typically comprises <1% of total storage but may have outsized biological activity owing to its ability to freely diffuse into cells. In the liver, VitD is converted into 25-hydroxy VitD [25(OH)D] by 1-a hydroxylase, which is primarily a CYP2R1-dependent process [8]. Of interest, VitD as 25(OH)D tends to be sequestered in adipose tissue and distributed in muscle in addition to serum [25,26]. While it is still under debate if free 25(OH)D–unbound to a carrier protein is a better method for measuring VitD status, the current clinical standard remains total serum 25(OH)D, which has a half-life of about one month [10]. While serum 25(OH)D is also generally used for research purposes, this is typically differentiated by form (VitD2 versus VitD3), with 25(OH)D3 being more efficacious due to its stronger affinity for the VitD receptor, and more is discussed below.

Ultimately, 25(OH)D is transformed into the biologically active form of VitD–1,25 di-hydroxy VitD [1,25(OH)_2_D], also known as calcitriol–by CYP27B1, primarily in the kidneys, skin keratinocytes, and immune cells (Figure 1). In contrast to the up to one month half-life of 25(OH), the half-life of the activated 1,25(OH)_2_D is only 4 h [10]. Further, it has been shown that the half-life of 25(OH)D2 is less than that of 25(OH)D3: just 13.9 ± 2.6 days for 25(OH)D2 versus 15.1 ± 3.1 days for 25(OH)D3 in a small cohort of young, healthy men [27]. Thus, supplementation with VitD2 is likely to remain in the system of a given individual for a shorter period of time than VitD3. VitD activation was historically thought to occur only in the kidneys, but it has been shown that skin keratinocytes and immune cells are also prodigious activators of 25(OH)D. In the kidneys, VitD primarily regulates calcium and phosphorus homeostasis while it functions as an immunomodulatory hormone in the immune system. Importantly, the conversion of 25(OH)D to calcitriol is influenced by several additional pathways including IGF-1, FGF23, and PTH [16]. Some 1,25(OH)_2_D enters cells via passive diffusion while most transportation of VitD metabolites requires facilitated diffusion via carrier proteins [28]. The kidneys can recognize 25(OH)D as a result of the megalin/cubilin complex with VDBP, enabling conversion to 1,25(OH)D in the kidneys. Once in the cell, VitD interacts with the VitD receptor (VDR), where it heterodimerizes with the retinoid X receptor (RXR), which then translocates into the nucleus and binds to VitD response elements (VDREs) in the promoter region of genes (Figure 1). Throughout the body, 1,25(OH)_2_D leads to autocrine and paracrine effects via the VDR/translocating into the nucleus to directly alter gene expression via VDREs with most cells expressing the VDR, allowing for a local fine tuning of gene expression via this mechanism [29]. Impressively, the VDR/RXR complex can regulate directly through the VDRE and indirectly (>1250 genes or 0.5–5% of the genome including but not limited to immune function, inflammation, and cardiovascular disease [30]).

### 2.2. Mechanism of Action of Vitamin D

The pancreas is no exception to the regulatory role of VitD. Interestingly, however, it not only expresses the VDR, but also all the elements of the VitD pathway, including 1-a hydroxylase and VDBP sites. Consonantly, deficiencies in VitD directly impair islet cell function, insulin receptor regulation, and regulation by RAAS elements [1]. In addition, the indirect action of VitD through intracellular calcium flux in L-type calcium channels mediating insulin release is strongly tied to 25(OH)D concentration. Similar effects are noted through the inverse regulation of pathological drivers of insulin resistance such as skeletal muscle adiposity in the context of fatty acid metabolism, a process controlled by peroxisome proliferator-activated receptor delta (PPARδ), which is a target of 25(OH)D [31,32]. Observation data have confirmed these findings with subjects in the lowest versus highest tertiles/quartiles of 25(OH)D facing outsized T2DM risk, where those in the highest quartiles enjoy an order of 30–70% relative risk reductions [33,34]. Persuasively, mendelian randomization analysis has similarly demonstrated reductions in T2DM risk in subjects with genetically predicted higher VitD status [35].

### 2.3. Factors Affecting Vitamin D Supplementation Efficacy

The bioavailability of VitD after supplementation is dependent on a number of factors, including the form of VitD, baseline VitD status (sigmoidal response curve), adiposity, age, obesity, kidney function, calcium status, and polymorphisms in the coding alleles for 1-a hydroxylase activity (CYP2R1), CYP27B1, and the VDBP. That certain subjects are more likely to see benefit from VitD therapy in RCTs is uncontested. With respect to VitD supplementation, it is known that VitD3 has a higher affinity for the VDBP in comparison with VitD2, reducing the comparative efficacy of VitD2 by approximately one third [36]. Further, supplementation with VitD3 has been shown to increase serum 25(OH)D more than the same dose of VitD2, with a systematic review finding an 8.08 ng/mL (20.19 nmol/L) difference [25].

In addition to the form of VitD that is delivered for supplementation, the baseline status of each individual dictates the dose–response as VitD supplementation exhibits a sigmoidal dose–response curve [37]. Therefore, those with the lowest and highest baseline VitD status will exhibit a blunted response to the same dose as those with more moderate VitD status at baseline. This alone is reason to perform titrate supplementation with periodic blood tests rather than assigning a specific dose based on a clinical presentation or diagnosis; however, there is significant interindividual variation even beyond this [38].

With respect to the aforementioned factors, adiposity (often approximated with BMI) has been explored as a predominant risk factor for insufficiency and the attenuated response to VitD intervention (Figure 2). At one time, there was hope that VitD could actually reduce adiposity and serve as an obesity therapy, but this has since been largely abandoned. In a Mendelian randomization analysis of multiple cohorts, Vimaleswaran et al. found only a small effect of VitD status on adiposity; however, with a 10% increase in BMI, a 4% decrease in serum 25(OH)D concentration was observed [26]. More recently, among prediabetic subjects randomized to 2000 IU VitD3, Pittas et al. noted that subjects with a BMI < 30 were less likely to convert to T2DM [39]. Such observational findings have been echoed by multiple meta-analyses documenting similar heterogeneity in response to therapy. Adipose tissue can profoundly sequester fat soluble VitD; in addition, 25(OH)D concentrations in adipose tissue have been documented during bariatric surgery and carries implications for intervention [40]. In keeping with this, 22–71% of subjects presenting for bariatric surgery have been shown to be VitD deficient at baseline (65–93% insufficient) [41]. Moreover, previous authors have estimated that subjects with elevated BMI required 2–3x the dose of VitD3 to achieve sufficient status [42]. Ekwaru et al. showed that BMI altered the trajectory of the dose–response to VitD supplementation with sharper response in underweight patients and blunted responses in overweight patients, suggesting daily requirements ranging from 28–1663 IU based on BMI category alone [43]. One study suggests an equation for determining a tailored VitD dose recommendation in patients with obesity: Additional daily vitamin D3 = [weight (kg) × desired change in 25(OH)D] − 10 [44].

However, the mechanisms governing these phenomena are in fact bidirectional and include the influence of VitD on leptin, modulating satiety and fullness, as well as the contribution of adipocyte tissue to the repression of the key CYPs in the VitD lifecycle—including CYP27B1 and CYP2J2—enzymes chiefly responsible for the production of active VitD [42,45,46,47,48]. In previous modeling studies, for each 1% of body mass, it has been reported that VitD increases by 0.7 nmol [49]. In fact, in an analysis of bariatric patients undergoing sleeve gastrectomy, Muzaffer et al. reported a significant decrease in the prevalence of VitD deficiency from 26.8% to 0.3%; this is likely due to the release of VitD from reducing adipose tissue with weight loss [50]. Impressively, this effect was sustained even in the absence of supplementation post-procedure. These results support observations from previous authors documenting the so-called “volume dilution” problem of VitD in the setting of those struggling with obesity. As the total available volume of VitD stores increases, 25(OH)D is inaccessible for activation—in other words, VitD is sequestered in adipose tissue and not bioavailable. Thus, the contribution of excess adiposity to failures in intervention cannot be excluded as explanatory, especially in the realm of T2DM prevention.

### 2.4. Genetic Variants and Epigentics

The most fundamental genetic variants that affect VitD status are those related to melanin concentration in the skin. However, this is not known to affect response to supplementation directly. Instead, the genetic and epigenetic variations related to VDR binding motifs and response elements, which yield the pleiotropic effects of VitD, appear to play a role and have been linked to numerous inflammatory diseases, including autoimmunity and cancer [51]. Similarly, the response to VitD supplementation has been shown to have three responder types—low, mid, and high—which are independent of serum 25(OH)D, with approximately 25% of the population being a low-responder type and thus requiring higher doses of VitD to reach similar VitD status as determined by serum 25(OH)D [52,53,54,55]. The VitD responder type appears to be a fixed, intrinsic characteristic; however, what specific genomic and epigenomic alterations lead to each responder type has yet to be discovered [55]. Therefore, screening for genetic or epigenetic variations is not currently possible; instead, supplementation dosing must be titrated for each individual based on VitD status as measured by serum 25(OH)D.

### 2.5. Possible Supplementation Regimens

There is a broad consensus in the literature regarding the need for titrating VitD3 supplementation to serum 25(OH)D—allowing the VitD status to determine the optimal dose rather than setting a dose for certain conditions. The personalized response to VitD supplementation necessitates this. However, clinicians can find this frustrating in terms of where to start with an individual patient. In a small study by Aloia et al., 138 patients were given 6 months of VitD3 supplementation with dose adjustments at 8 and 16 weeks to determine the dose required for all patients to reach sufficiency (30 ng/mL, 75 nmol/L) [56]. Applying the model created in these 138 patients to the third National Health and Nutrition Examination Survey (NHANESIII), Aloia et al. estimated two dosing regimens:(1)For 25(OH)D > 22 ng/mL (55 nmol/L): 3800 IU vitamin D3 daily;(2)For 25 (OH)D < 22 ng/mL (55 nmol/L): 5000 IU vitamin D3 daily.

These can act as guidelines for where to start dosing VitD3 when the baseline serum 25(OH)D status is known.

## 3. Sex-Specific Response

### 3.1. The Role of Estrogens

The effects of VitD appear different between the sexes, which is largely a function of the role of estrogens. Estradiol (E2), the estrogen that is created endogenously in the ovaries, decreases the expression of CYP24A1, the cytochrome P450 component of the 25-hydroxyvitamin D(3)-24-hydroxylase enzyme, which inactivates vitamin D ([57], p. 3). In short, this relationship leads to a greater accumulation of VitD in fat (sequestration) and general pools (serum and muscle). Greater pools of VitD increase the anti-inflammatory ability of VitD, improving immune function in women, and consequently decreasing the risk for certain autoimmune conditions. Importantly, this relationship is absent in men. The role of estrogen in increasing VitD is further supported by studies that identify a correlation between 25(OH)D concentration and the use of contraceptives containing estrogen [58]. Specifically, research has shown that even VitD-deficient women see significant increases in their VitD status following the introduction of these estrogen-containing contraceptives. There is currently no research that considers the role of fluctuations in estrogen across a lifespan (i.e., how menopause may impact VitD in women) [57].

### 3.2. The Influence of Sex on Adiposity

Conflicting research suggests that VitD deficiency is higher among women of all BMI classifications when compared to men with similar weight profiles [59]. The authors of this study specifically hypothesize that the higher prevalence of deficiency among women may be a function of the greater adiposity found in women across BMI classifications, which would lead to a greater sequestration of VitD in this adipose tissue. These findings directly challenge the assumptions of previously published studies [16,57]. The results of Muscogiuri et al. are challenged by another study similarly comparing serum 25(OH)D among men and women with obesity and T2DM. Here, we see that both men and women with T2DM present with VitD deficiency, but the authors identify significantly lower serum VitD among men [60]. The authors note that VitD is a fat soluble hormone but argue that increased adiposity does not result in sequestration or a reduced bioavailability of the vitamin. They further posit that women experience higher VitD status given their larger stores of adipose tissues, even when compared against men of similar BMI classifications. However, this is contrary to much of the literature on VitD and adipose tissue, including the many studies that show the VitD status in individuals undergoing bariatric surgery improves with weight loss (reduced adipose tissue), which this lab has previously discussed [32,40,61]. Further, it is generally accepted that VitD is not stored like other fat soluble vitamins, which are stored in the liver.

### 3.3. Sex-Specific Response to Interventions

Additional research also suggests that women with higher 25(OH)D status and prediabetes show a more positive metabolic profile overall, including lower total cholesterol and higher HDL than among similar cohorts of men. This same study found a negative relationship between 25(OH)D concentration in men and their total cholesterol, fatty liver index, and insulin response [62]. Sex differences have also been observed in high-dose VitD supplementation among critically ill men and women in a randomized placebo-controlled trial. In this study, women received higher doses of VitD than their male counterparts but showed significantly lower VitD absorption than males [63]. Additional research has shown sex-specific differences in coagulation and blood lipids during VitD intervention. For example, a study reviewing the impact of VitD levels following increased sun exposure and diet change among overweight adults incidentally found a distinct sex-specific difference in blood coagulation enrichment [64]. Another study observing a low-fat dietary intervention for increasing VitD status among otherwise metabolically health adults found that blood lipids decreased among the women across the duration of the study, while men did not see such a protective outcome [65]. In sum, the evidence related to the sex-specific differences in VitD kinetics/dynamics and how this affects function are conflicting and largely based upon observational research [66]. In order to better understand these biological differences, robust and randomized clinical trials (RCTs) are necessary.

## 4. Type 2, Oxidative Stress, and Autoimmunity

T2DM is a heterogeneous metabolic derangement that can be understood as the product of a coordinated cross-talk between insulin resistance, oxidative stress, and overlying auto-immune pathology that distinctly accelerates and, in some cases, may define the conversion of cases of T2DM-like pathology. The pleiotropic effects of VitD on enzymatic stress and autoimmunity situate the vitamin as a unique non-pharmacologic lever in the setting of T2DM treatment (Figure 3).

Chronic oxidative stress has been proposed as a driver of T2DM and insulin resistance [67,68] States of hyperglycemia are definitionally toxic: hence, the term “glucotoxicity”, which commonly refers to the production of the reactive oxygen species (ROS) that are produced as a consequence of hyperglycemic metabolic states. Oxidative stress in this context can be induced through a variety of mechanisms, including changes in the ratio of NADH to NADPH—the former actually favors ROS—as well as the formation of advanced end products (AGEs) and protein kinase-C mediated NADPH oxidase dysfunction [67]. Hyperglycemic states induce characteristic patterns of DNA damage, peroxidation, and targeted stress to various tissues, including islet beta cells directly, which have been demonstrated to contribute to insulin-resistance-mediated pathology [67,69]. VitD has been observed to fortify various parameters of oxidative stability, including its ability to downregulate the development of AGEs and replenish stores of glutathione (GSH), thereby decreasing ROS formation and improving glucose control. In an RCT of patients with T2DM, Gu et al. demonstrated that VitD inversely correlated with markers of oxidative stress—such as GSH, monocyte chemoattractant protein 1 (MCP-1), and IL-8—in comparison with healthy control groups [34]. Supplementation with VitD rescued this phenotype and facilitated a significant decrease in markers of oxidative stress while increasing GSH in the T2DM group [34]. In subsequent RCTs, VitD was able to similarly reduce markers of oxidative stress, which were associated with improvements in A1c and glucose homeostasis [70,71]. Taken together, these trials suggest that oxidative stress likely contributes to the pathogenesis and severity of T2DM, and that there may be a role for VitD in minimizing oxidative stress both in the pathogenesis of and treatment of T2DM. VitD exerts profound effects on the immune system, augmenting clonal cell populations, inflammatory species, and remodeling the landscape of the innate and adaptive immune systems alike. Overall, VitD promotes homeostasis and immune surveillance while also allowing for a robust immune response in the presence of a pathogen or rogue cell (autoimmunity, cancer). For instance, VitD enhances tolerance-inducing Treg populations, depresses B cell proliferation, reduces LPS-induced cytotoxicity, skews toward a Th2 response (over Th1, Th17, etc.), and decreases numerous pro-inflammatory cytokines (MCP-1, IL-1, IL-2, IFN-γ, TNF-α, IL-17, IL-21, and iNOS) while boosting anti-inflammatory cytokines (IL-4, IL-10, and IL-13) [29,61,62,63]. Even in the absence of T2DM, excess adiposity (roughly estimated by BMI) elevates serum inflammatory markers (IL-6, TNF, and MCP-1) [67,72]. Moreover, pro-inflammatory cytokines activate suppressors of cytokine signaling (SOCs), elements known to induce insulin resistance and contribute to the susceptibility to T2DM [67]. Further, anti-inflammatory, insulin-sensitizing adipokines, such as adiponectin, are notably absent in individuals with overweight and diabetes [67,72]. Of note, previous authors have not only linked serum adiponectin with VitD status, but have also shown improvement following supplementation [73,74,75,76]. In this way, inflammation is secondary to excess adiposity in the context of obesity, and T2DM is associated with VitD status; importantly, it may also improve these markers with supplementation.

The pathogenesis of T2DM does not, by definition, include auto-immunity—in fact, the presence of auto-antibodies is classically suggestive of T1DM. However, several large, diverse and representative cohorts of subjects diagnosed with T2DM have called into question the role of auto-immunity in subjects diagnosed with T2DM—perhaps pointing toward a more nuanced approach in the work up of some T2DM patients. In both relative and absolute numbers, the presence of immune-mediated destruction in those diagnosed with T2DM is significant, with a median of about 10% of subjects at large. Furthermore, the representative “T2DM cohorts” tested positive for one or more auto-antibodies, including glutamate decarboxylase (GAD), islet cytoplasmic antibodies (ICA), insulinoma-associated protein 2 (IA-2A), or zinc transporter 8 (ZnT8) [77,78,79,80,81,82]. In general, such findings in the setting of T2DM-like clinical presentation are given a diagnosis of latent autoimmune diabetes (LADA) with the pre-test probability being enhanced by criteria such as age > 30–35, BMI 18–25, and low or normal C-Peptide. We cautiously remark that in those subjects diagnosed with T2DM, the burden of auto-immunity may be more substantial than is commonly understood. Several prominent population studies support this approach. The UK Prospective Diabetes Study (1997) tested 3672 white patients with T2DM, aged 25 to 65 years, for autoantibodies (GAD, ICA) [77,83]. In newly diagnosed subjects, 10% tested positive for one or both [77]. In a smaller Pittsburgh cohort of the Cardiovascular Health Study (2000), of patients newly diagnosed with T2DM (n = 196), 12% were positive for autoantibodies (GAD, IA-2A) [80]. Similarly, a 2013 European cohort of 6156 patients reported a positivity rate of >9.7% for at least one autoantibody (GAD, IA-2A, ZnT8A), where it was found that younger subjects that were leaner, insulin-dependent, and female were more likely to be diagnosed with LADA [81]. More recently, a Norwegian HUNT Study (2018) documented >7–8% positivity (GAD) in 2002 individuals recently diagnosed with diabetes; a younger age significantly increased the risk of autoantibody positivity with heterogeneity in weight [79]. Altogether, there is likely a distinct subpopulation of those misdiagnosed with T2DM that have clinical markers of autoimmunity and, therefore, islet destruction that contributes to insulin dependence; this means that the correct diagnosis is actually LADA.

As an immunomodulatory hormone, the role of VitD in the primary prevention of autoimmune-associated disease (AAD) has demonstrated compelling results. The VITAL Study documented a significant reduction in AAD incidence (22%) in 25,768 subjects randomized to 2000 IU VitD3 with or without omega-3 fatty acids [78]. Furthermore, in a trial of 38 subjects recently diagnosed with type 1 diabetes, Gabbay et al. reported that, in those randomized to 2000 IU VitD per day, only 18.5% progressed to undetectable C-peptide, whereas this effect was only noted in 62.5% of controls [83]. In addition, numerous population-level analyses, spanning decades, have provided observational corroboration of these findings with those subjects with higher VitD status at lower risk for AAD. Thus, that VitD may obviate the development of specific cases of T2DM is provocative and should be explored by subsequent active screening for leaner, younger T2DM phenotypes that may preferentially benefit from the anti-inflammatory character of VitD.

## 5. Future Perspectives and Conclusions

Research affirms that there are relationships between VitD sufficiency, autoimmune function, and the prevention of disease; however these relationships are not perfectly understood. Additional research is crucial as there remain many questions unanswered related to VitD status and conditions such as T2DM. Future research must be conducted to better understand the potentially spurious relationships between VitD status, supplementation, sun exposure, health behaviors, and the diagnosis and management of T2DM. Public health practice can greatly benefit from a better understanding of the mechanisms by which we can reliably increase VitD status and how this can be used to develop education and improve health behaviors.

## Figures and Tables

**Figure 1 ijerph-20-06187-f001:**
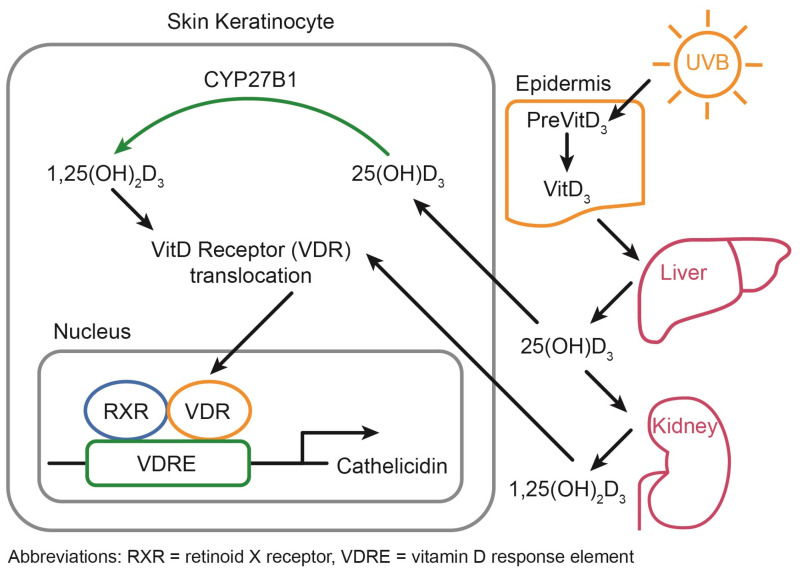
Vitamin D Metabolism.

**Figure 2 ijerph-20-06187-f002:**
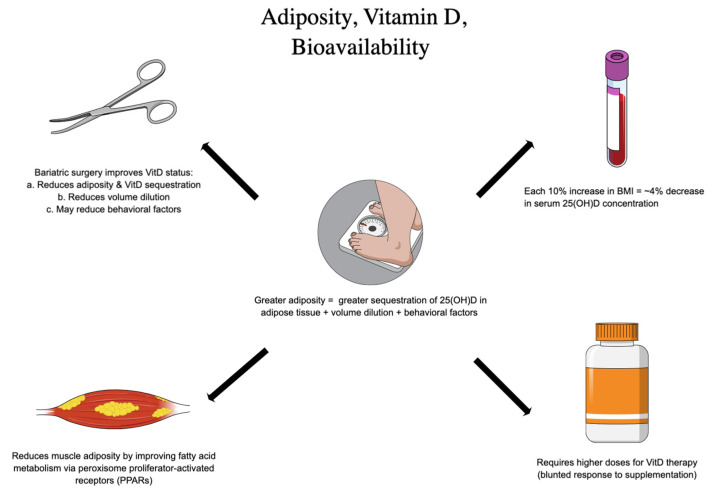
Adiposity and Vitamin D Bioavailability.

**Figure 3 ijerph-20-06187-f003:**
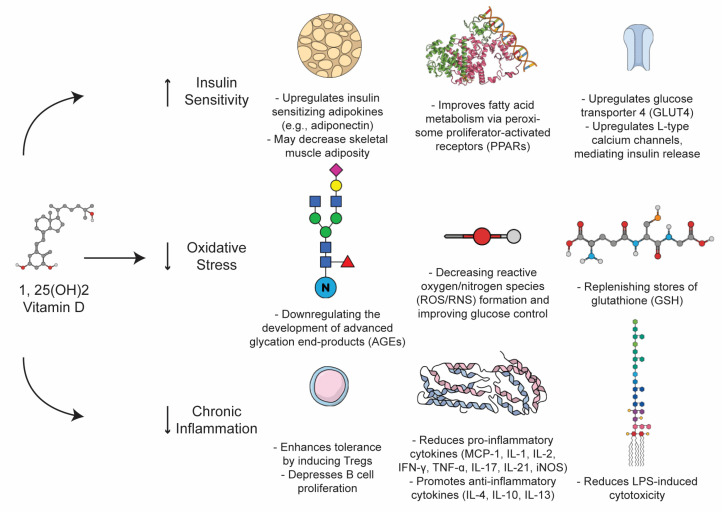
The Role of Vitamin D in Insulin Sensitivity and Inflammation.

## Data Availability

Data sharing not applicable.

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
