# Peer review of "Clarifying the Heterogeneity in Response to Vitamin D in the Development, Prevention, and Treatment of Type 2 Diabetes Mellitus: A Narrative Review"

_ijerph, 2023, doi:10.3390/ijerph20126187_

Round 1

Reviewer 1 Report

The title of the review specifies "Clarifying the role of vitamin D in the development, prevention, and treatment of type 2 diabetes mellitus: a narrative reviews "

However, in the paragraph "Type 2, Oxidative Stress, and Autoimmunity", (starting from line 220) a little confusion is made between type 2 diabetes, LADA and finally Type 1 diabetes.

It must be clarified that in type 2 diabetes there are no antibodies and that the presence of antibodies refers to LADA or type 1 diabetes (especially if it refers to young subjects)

The discussion in this paragraph seems to focus on vitamin D and diabetes in general rather than just type 2 diabetes and needs to be changed to specify autoimmunity separately (additional paragraph?)

Author Response

Thank you for your feedback. You raise the important point that the burden of autoimmunity in T2DM is not well known outside the field, and we have added language to clarify this.

Reviewer 2 Report

There are numerous papers and over 100 review articles on this topic of vitamin D and type 2 diabetes. The mentioned article does not bring anything new to this field, and I believe that this manuscript is not adequate for further publication.

Author Response

Thank you for your perspective. We believe that this provides not just an update but a different perspective than previous reviews, particularly as an editorial for this special issue.

Further, the other 6 reviewers disagree and include comments such as:

  • “the review seems to compile some of the latest clinical challenges that the field is facing”
  • “well written clear report of the current state of the literature from the researchers perspective.  This topic is highly relevant at this time.”
  • “well written and provides useful insights into the topic.”

Reviewer 3 Report

Review Response

In this review, authors explored the likely reasons of heterogeneity in response to VitaminD therapy in those at risk for and diagnosed with T2DM. They mention some of the factors driving this variability to supplementation such as bioavailability, sex-specific response, oxidative stress, autoimmune pathology etc. and propose to study distinct populations for future intervention with VitD. Overall, the review seems to compile some of the latest clinical challenges that the field is facing. However, I have some major suggestions as follows:

1.     The review seems to be full of text with no figures. Readers get the interest to read an article when they see the figures explaining the article in the simplest way. It is recommended to have a pictorial representation of how Vitamin D status is regulated including its precursors and active forms across different human tissues. Also, within a cell, giving few examples of VitD response elements (VDREs) genes regulated by VDR/RXR complex within pancreas in the figure is highly encouraged. 

2.     Although, authors propose to study distinct populations for future intervention with VitD supplementation which is impossible without studying the population-specific genetic architectures. Authors seem to ignore the role of genetic variations in regulating basal vitamin D level that are population specific. Genetic variations identified for Vitamin D levels across different ethnic populations can be used to design population-specific genetic testing methods where a clinician can draw conclusions based on a genetic testing and timely clinical measurement of vitamin D levels of the same individual. Sometimes, what clinicians think a deficiency of vitamin D level can be the normal level of vitamin D for a specific population if the levels keep constant for many years in individuals harboring specific genotype and perhaps, they may not need supplementation. Such conclusions can only be drawn when we have combined genetic and clinical data for the individual samples. It is imperative to add the importance of genetic variants in designing population-specific supplementation strategies in the revised version.

3.     Authors emphasize the deficiency of vitamin D levels in T2DM individuals. However, what are the toxicity effects of elevated vitamin D levels say through supplementation? Are there any reports pertaining to this? And how optimal supplementation levels of vitamin D are chosen that is sufficient but not toxic? This can be also the challenge for clinicians.

4.     Authors mention in line numbers between 172 to 174 that “As the total available volume of VitD stores increases, 25(OH)D is in-accessible for activation–in other words, VitD is sequestered in adipose tissue and not bi-available”. Have there been studies that report the ways to reverse this process? In other words, can the stored vitamin D in adipose tissue be re-converted to active forms via exercise or other physical interventions? Since, this can be helpful to prevent the individuals from progressing towards T2DM. These alternative ways to elevate the levels of vitamin D when the optimum levels of exogenous supplementation are not known can be discussed in the article.

Author Response

In this review, authors explored the likely reasons of heterogeneity in response to VitaminD therapy in those at risk for and diagnosed with T2DM. They mention some of the factors driving this variability to supplementation such as bioavailability, sex-specific response, oxidative stress, autoimmune pathology etc. and propose to study distinct populations for future intervention with VitD. Overall, the review seems to compile some of the latest clinical challenges that the field is facing. However, I have some major suggestions as follows:

  1. The review seems to be full of text with no figures. Readers get the interest to read an article when they see the figures explaining the article in the simplest way. It is recommended to have a pictorial representation of how Vitamin D status is regulated including its precursors and active forms across different human tissues. Also, within a cell, giving few examples of VitD response elements (VDREs) genes regulated by VDR/RXR complex within pancreas in the figure is highly encouraged. 

Thank you for your feedback. Other reviewers have also requested figures, and we have added three figures to the manuscript. One addresses the basic metabolism of vitamin S at its primary site, the skin, as you suggested.

  1. Although, authors propose to study distinct populations for future intervention with VitD supplementation which is impossible without studying the population-specific genetic architectures. Authors seem to ignore the role of genetic variations in regulating basal vitamin D level that are population specific. Genetic variations identified for Vitamin D levels across different ethnic populations can be used to design population-specific genetic testing methods where a clinician can draw conclusions based on a genetic testing and timely clinical measurement of vitamin D levels of the same individual. Sometimes, what clinicians think a deficiency of vitamin D level can be the normal level of vitamin D for a specific population if the levels keep constant for many years in individuals harboring specific genotype and perhaps, they may not need supplementation. Such conclusions can only be drawn when we have combined genetic and clinical data for the individual samples. It is imperative to add the importance of genetic variants in designing population-specific supplementation strategies in the revised version.

Thank you for your feedback. Indeed, we did not include much on genetic/epigenetic variants in this manuscript, as we, originally, felt this was too detailed and may take away from the key points. However, we can see that a small section on this is necessary and have added this.

  1. Authors emphasize the deficiency of vitamin D levels in T2DM individuals. However, what are the toxicity effects of elevated vitamin D levels say through supplementation? Are there any reports pertaining to this? And how optimal supplementation levels of vitamin D are chosen that is sufficient but not toxic? This can be also the challenge for clinicians.

Toxicity is extremely rare. It has not been observed from sunlight.1 When it comes to vitamin D supplementation, this has almost always occurred in cases of manufacturing issues with products that contain many times what they are supposed to and, even still, this takes months of use. Therefore, this is not a chief clinical concern, especially when working with patients in states of deficiency or insufficiency where the supplementation is being titrated to a certain VitD status (well below toxicity).

Potential side effects from vitamin D supplementation include hypercalcemia and have been observed with intakes above 50,000 IU daily, an extraordinarily high dose.2 Given that most supplementation regimens fall under 10,000 IU daily, toxicity is very unlikely.3-5

  1. Volmer DA, Mendes LR, Stokes CS. Analysis of vitamin D metabolic markers by mass spectrometry: Current techniques, limitations of the "gold standard" method, and anticipated future directions. Mass Spectrom Rev. 2015;34(1):2-23.
  2. Holick MF. Vitamin D deficiency. N Engl J Med. 2007;357(3):266-281.
  3. Vieth R. Vitamin D supplementation, 25-hydroxyvitamin D concentrations, and safety. Am J Clin Nutr. 1999;69(5):842-856.
  4. Heaney RP, Davies KM, Chen TC, Holick MF, Barger-Lux MJ. Human serum 25-hydroxycholecalciferol response to extended oral dosing with cholecalciferol. Am J Clin Nutr. 2003;77(1):204-210.
  5. Vieth R, Chan PC, MacFarlane GD. Efficacy and safety of vitamin D3 intake exceeding the lowest observed adverse effect level. Am J Clin Nutr. 2001;73(2):288-294.
  6. Authors mention in line numbers between 172 to 174 that “As the total available volume of VitD stores increases, 25(OH)D is in-accessible for activation–in other words, VitD is sequestered in adipose tissue and not bi-available”. Have there been studies that report the ways to reverse this process? In other words, can the stored vitamin D in adipose tissue be re-converted to active forms via exercise or other physical interventions? Since, this can be helpful to prevent the individuals from progressing towards T2DM. These alternative ways to elevate the levels of vitamin D when the optimum levels of exogenous supplementation are not known can be discussed in the article.

There are no studies showing this process can be reversed, per say; however, we discussed the role of weight loss on improving vitamin D status in our lab’s previous work. We did not feel this was highly relevant to this manuscript and would detract from the main message. Therefore, we only discussed this briefly in the sections on Factors Affecting Vitamin D Supplementation Efficacy and The Influence of Sex on Adiposity.

Reviewer 4 Report

Please see the attachment file.

Author Response

Overall, the paper is well written. Here are some recommendations for improvement.

1) Please do not abbreviate any vitamin D species. Just use as vitamin D or ergocalciferol or cholecalciferol.

This is common throughout the literature and in many of our lab’s previous publications. This is a personal preference, and we would prefer to continue with our lab’s style.

2) Line 32 to 33: Vitamin D concentration of less than 20 ng/dL is not a universally accepted standard. There is a controversy shrouds this concept. The IOM recommendations vs. Endocrine Society recommendations.

Thank you for your perspective. This is true, however, this manuscript does not focus on this controversy, so we saw it as beyond the scope. Now, we have added more information regarding this as background.

3) Introduction: The relationship between serum vitamin D and the risk of type 2 diabetes has been well established. I am surprised to see a lack of synthesis of data from epidemiological evidence. I would recommend that authors create a subheading related to this topic and give a balanced opinion on the existing literature. Here are some references to start with.

  1. Serum Vitamin D Concentration ≥75 nmol/L Is Related to Decreased Cardiometabolic and Inflammatory Biomarkers, Metabolic Syndrome, and Diabetes; and Increased Cardiorespiratory Fitness in US Adults. Nutrients. 2020;12(3):730
  2. Association Between Plasma Vitamin D2 and Type 2 Diabetes Mellitus. Front Endocrinol. 2022 Jun 1;13:897316. doi: 10.3389/fendo.2022.897316
  3. Wang N, Wang C, Chen X, et al. Vitamin D, prediabetes and type 2 diabetes: bidirectional Mendelian randomization analysis. Eur J Nutr. 2020;59(4):1379-1388.

Thank you for this recommendation. We agree that a brief overview of epi research is valuable and have added some of this background in the introduction to our manuscript (see references 85-87).

4) Recently (last month) this paper was published. Please include this in your review.

Vitamin D and Risk for Type 2 Diabetes in People With Prediabetes : A Systematic Review and Meta-analysis of Individual Participant Data From 3 Randomized Clinical Trials. Ann Intern Med. 2023;176(3):355-363.

Thank you for bringing this resource to our attention. We have added it to our reference list and cited within the manuscript.

5) Line 90: 25(OH)D does not have a half-life of 1 month. It is half of that. Please see this paper.

25(OH)D2 half-life is shorter than 25(OH)D3 half-life and is influenced by DBP concentration and genotype. J Clin Endocrinol Metab. 2014 Sep;99(9):3373-81. doi: 10.1210/jc.2014-1714

The half-life of 25(OH)D is only about 15 days are so although it may vary depending on age and overall health. This fact needed to be corrected.

Thank you for the feedback and the reference. We have clarified this section.

6) Line 47-108: This is simply a very basic physiology of vitamin D. I would suggest reducing this by half. Give references for very basic science.

Thank you for your perspective. We do not disagree. The other 6 reviewers do not seem to agree with removing this basic metabolism background; in fact, asking us to expand our background. Also, we think that many will not have this prior knowledge, and, thus support leaving this section intact.

Reviewer 5 Report

The narrative review entitled  Clarifying the Role of Vitamin D in the Development, 2 Prevention, and Treatment of Type 2 Diabetes Mellitus was a well written clear report of the current state of the literature from the researchers perspective. 

This topic is highly relevant at this time.

Author Response

Thank you for your review.

Reviewer 6 Report

This is a perspective piece aiming to discuss drivers of heterogeneity in the findings of studies investigating the association between vitamin D and T2D. The manuscript is well written and provides useful insights into the topic.

I have some suggestions for improvement:

1.     Abstract: The abbreviation VitD is defined in line 11, while it appears for the first time in line 8

2.     When the authors refer to sufficiency cut-off points, I think it is important to clarify that these cut-offs have been established in relation to skeletal outcomes. To observe improvements in diabetes-related outcomes, one should probably go above 30 ng/ml. For example, a small RCT proved that subjects with prediabetes achieving 25(OH)D concentrations > 32 ng/ml experienced a further reduction in fasting glucose compared to those reaching levels <32 ng/ml (https://doi.org/10.1080/17512433.2022.2043153). Please also consider this publication, showing improvements in IR for concentrations greater than 32 ng/ml: https://doi.org/10.1017/s0007114509992017

3.     Regarding the gender-specific response to vitamin D supplementation, it may be useful to comment on the findings of a small study that demonstrates that vitamin D status is related to a distinct response to blood lipid levels between men and women following a low-fat diet (https://doi.org/10.1016/j.jsbmb.2021.105903). Furthermore, given that T2D is a state of increased risk of thrombosis, consider this study showing sex-specific vitamin D effects on blood coagulation among overweight individuals (https://doi.org/10.1111/eci.12688)

4.     A figure summarizing the role of vitamin D in T2D would help the reader to put into context the data discussed in the manuscript.

5.     Finally, I think the title is too general and thus does not accurately reflect the content of the manuscript. In any case, the association between vitamin D and T2D is a vast field. I would suggest that the authors 'narrow' the title and focus on the topics discussed, eg, sex-specific response, autoimmunity, etc.

Author Response

This is a perspective piece aiming to discuss drivers of heterogeneity in the findings of studies investigating the association between vitamin D and T2D. The manuscript is well written and provides useful insights into the topic.

I have some suggestions for improvement:

  1. Abstract: The abbreviation VitD is defined in line 11, while it appears for the first time in line 8

 Thank you for catching that. We have corrected this.

  1. When the authors refer to sufficiency cut-off points, I think it is important to clarify that these cut-offs have been established in relation to skeletal outcomes. To observe improvements in diabetes-related outcomes, one should probably go above 30 ng/ml. For example, a small RCT proved that subjects with prediabetes achieving 25(OH)D concentrations > 32 ng/ml experienced a further reduction in fasting glucose compared to those reaching levels <32 ng/ml (https://doi.org/10.1080/17512433.2022.2043153). Please also consider this publication, showing improvements in IR for concentrations greater than 32 ng/ml: https://doi.org/10.1017/s0007114509992017

Excellent point. We were too focused on brevity originally. We have expanded this section to discuss this and included the references you suggested. Thank you.

  1. Regarding the gender-specific response to vitamin D supplementation, it may be useful to comment on the findings of a small study that demonstrates that vitamin D status is related to a distinct response to blood lipid levels between men and women following a low-fat diet (https://doi.org/10.1016/j.jsbmb.2021.105903). Furthermore, given that T2D is a state of increased risk of thrombosis, consider this study showing sex-specific vitamin D effects on blood coagulation among overweight individuals (https://doi.org/10.1111/eci.12688)

 Thank for these resources and the recommendations. We agree that this material is relevant and have spoken briefly to both studies in the “Sex-Specific Response” section of the manuscript.

  1. A figure summarizing the role of vitamin D in T2D would help the reader to put into context the data discussed in the manuscript.

Thank you for your feedback. Other reviewers have also requested figures, and we have added three figures to the manuscript. One addresses the role of vitamin D in T2DM, as you suggested.

  1. Finally, I think the title is too general and thus does not accurately reflect the content of the manuscript. In any case, the association between vitamin D and T2D is a vast field. I would suggest that the authors 'narrow' the title and focus on the topics discussed, eg, sex-specific response, autoimmunity, etc.

Thank you for this helpful feedback. While we think our abstract clearly hit on the crux of the issue, the tile was, indeed, too general. We have updated this to be more focused:

Clarifying the Heterogeneity in Response to Vitamin D in the Development, Prevention, and Treatment of Type 2 Diabetes Mellitus: A Narrative Review 

Reviewer 7 Report

The manuscript under review explores potential drivers of heterogeneity in response to Vitamin D therapy in individuals at risk for and diagnosed with type 2 diabetes mellitus (T2DM). The authors examine factors such as bioavailability, sex-specific response, and autoimmune pathology and propose distinct populations for future intervention with Vitamin D.

Introduction:

Provide context for the contradictory findings in the literature: The authors mention that the literature on VitD supplementation in T2DM is complex and often contradictory, but they do not provide much context for why this might be the case. Adding a brief discussion of potential reasons for the contradictory findings (e.g., variability in study design, differences in populations studied) would help to frame the review and provide a rationale for the authors' approach.

Consider reorganizing the introduction for clarity: Currently, the introduction jumps around a bit between different topics (e.g., the importance of VitD status, the pleiotropic effects of VitD, potential drivers of heterogeneity in response to VitD therapy).

Section on Bioavailability & Response with Supplementation:

Use subheadings: The passage covers a lot of ground, and subheadings could help the reader to navigate through the different topics. For example, subheadings like "Mechanism of Action of Vitamin D", "Impact of Vitamin D on Insulin Resistance", and "Factors Affecting the Efficacy of Vitamin D Supplementation" could help to break up the text and make it more manageable for the reader.

Sex-Specific Response

The passage covers a range of topics related to sex-specific responses to vitamin D, such as the impact of estrogen and the relationship between vitamin D and adipose tissue. Organizing the information into subheadings would make it easier for readers to follow the main points and understand the structure of the passage.

Adding visual aids such as charts or diagrams can help readers understand complex relationships or trends mentioned in the passage, such as the relationship between vitamin D and adipose tissue or the impact of estrogen on vitamin D absorption.

Author Response

The manuscript under review explores potential drivers of heterogeneity in response to Vitamin D therapy in individuals at risk for and diagnosed with type 2 diabetes mellitus (T2DM). The authors examine factors such as bioavailability, sex-specific response, and autoimmune pathology and propose distinct populations for future intervention with Vitamin D.

Introduction:

Provide context for the contradictory findings in the literature: The authors mention that the literature on VitD supplementation in T2DM is complex and often contradictory, but they do not provide much context for why this might be the case. Adding a brief discussion of potential reasons for the contradictory findings (e.g., variability in study design, differences in populations studied) would help to frame the review and provide a rationale for the authors' approach.

Thank you for this excellent feedback. We have expanded upon this, which we think strengthens the introduction.

Consider reorganizing the introduction for clarity: Currently, the introduction jumps around a bit between different topics (e.g., the importance of VitD status, the pleiotropic effects of VitD, potential drivers of heterogeneity in response to VitD therapy).

This section has been significantly fleshed out at the request of other reviewers. We hope this has improved the clarity.

Section on Bioavailability & Response with Supplementation:

Use subheadings: The passage covers a lot of ground, and subheadings could help the reader to navigate through the different topics. For example, subheadings like "Mechanism of Action of Vitamin D", "Impact of Vitamin D on Insulin Resistance", and "Factors Affecting the Efficacy of Vitamin D Supplementation" could help to break up the text and make it more manageable for the reader.

Excellent suggestion. Thank you. We have incorporated subheadings into the manuscript.

Sex-Specific Response

The passage covers a range of topics related to sex-specific responses to vitamin D, such as the impact of estrogen and the relationship between vitamin D and adipose tissue. Organizing the information into subheadings would make it easier for readers to follow the main points and understand the structure of the passage.

Thank you for this helpful suggestion. We have added subheadings.

Adding visual aids such as charts or diagrams can help readers understand complex relationships or trends mentioned in the passage, such as the relationship between vitamin D and adipose tissue or the impact of estrogen on vitamin D absorption.

Thank you for your feedback. Other reviewers have also requested figures, and we have added three figures to the manuscript. One addresses the relationship between vitamin D and adiposity, as you suggested.

Round 2

Reviewer 2 Report

Dear editor, I stand by my comment and opinion.

Author Response

Thank you for your review.

Reviewer 3 Report

The manuscript seems improved after revision and can be accepted in present form after minor English spell checks. Figure resolution is also encouraged to improve.

Author Response

Thank you for your review.

Reviewer 4 Report

The authors have submitted a revised version of their review. Unfortunately, the authors have not incorporated several of my recommendations.

I have given a few references related to this topic to make the review a bit more balanced. They have not included those references.

One author's papers were over-referenced. 10% of references came from one author. That is a very disproportional representation of references from one author. 

Author Response

I have given a few references related to this topic to make the review a bit more balanced. They have not included those references.

Thank you for your feedback. We felt this was outside the scope of this paper, which was in line with other reviewers' suggestions to streamline this (in contrast to your feedback). However, we have added a brief mention of this now in the hope that this will satisfy all reviewers; thank you for understanding.

One author's papers were over-referenced. 10% of references came from one author. That is a very disproportional representation of references from one author.

Thank you for your review. Indeed, the Holick Lab is a prodigious leader in the vitamin D field, producing much of the seminal work in this field, which has relatively limited labs working in it. Thus, we must cite a good deal of work from the Holick Lab. However, we did intentionally work to include the work of other labs. The references reflect the state of the field.